# Anomalous enhancement of charge density wave in kagome superconductor CsV₃Sb₅ approaching the 2D limit

Boqin Song[1,10], Tianping Ying [2,10] ✉, Xianxin Wu[3,10], Wei Xia[4,5,10], Qiangwei Yin[6,10], Qinghua Zhang[2], Yanpeng Song[2], Xiaofan Yang[1], Jiangang Guo [2], Lin Gu [2], Xiaolong Chen[2], Jiangping Hu [2], Andreas P. Schnyder [7], Hechang Lei [6] ✉, Yanfeng Guo [4] ✉ & Shiyan Li [1,8,9] ✉

The recently discovered kagome metals AV₃Sb₅ (A = Cs, Rb, K) exhibit a variety of intriguing phenomena, such as a charge density wave (CDW) with time-reversal symmetry breaking and possible unconventional superconductivity. Here, we report a rare non-monotonic evolution of the CDW temperature ($T_{CDW}$) with the reduction of flake thickness approaching the atomic limit, and the superconducting transition temperature ($T_c$) features an inverse variation with $T_{CDW}$. $T_{CDW}$ initially decreases to a minimum value of 72 K at 27 layers and then increases abruptly, reaching a record-high value of 120 K at 5 layers. Raman scattering measurements reveal a weakened electron-phonon coupling with the reduction of sample thickness, suggesting that a crossover from electron-phonon coupling to dominantly electronic interactions could account for the non-monotonic thickness dependence of $T_{CDW}$. Our work demonstrates the novel effects of dimension reduction and carrier doping on quantum states in thin flakes and provides crucial insights into the complex mechanism of the CDW order in the family of AV₃Sb₅ kagome metals.

The geometric motif of the kagome lattice inherently leads to flat bands, Dirac points, and van Hove singularities (VHS). Depending on the electron filling, non-trivial topology and electronic correlations can emerge, which will induce a wealth of correlated phenomena encompassing quantum magnetism[1–4], charge or spin density wave[5–8], and superconductivity[7–9]. Generally, correlation will drastically grow and become prominent along with an enhanced density of states (DOS) at van Hove fillings when approaching the 2D limit, which may give rise to

unprecedented phenomena. Although many kagome magnets, including Mn₃Sn[10,11], TbMn₆Sn₆[12], and Co₃Sn₂S₂[13,14], have been identified, their strong interlayer couplings prevent a realization of 2D kagome lattices through an exfoliation process. It is therefore highly desirable to find an exfoliable kagome material and study thin flakes to explore intrinsic properties of 2D kagome lattices.

The recently discovered kagome metal AV₃Sb₅ (A = K, Rb, Cs)[15–18] has sparked a flurry of research interests, owing to possible

[1]State Key Laboratory of Surface Physics, Department of Physics, Fudan University, Shanghai 200433, China. [2]Beijing National Laboratory for Condensed Matter Physics, Institute of Physics, Chinese Academy of Sciences, Beijing 100190, China. [3]CAS Key Laboratory of Theoretical Physics, Institute of Theoretical Physics, Chinese Academy of Sciences, Beijing 100190, China. [4]School of Physical Science and Technology, ShanghaiTech University, Shanghai 201210, China. [5]ShanghaiTech Laboratory for Topological Physics, Shanghai 201210, China. [6]Laboratory for Neutron Scattering, and Beijing Key Laboratory of Optoelectronic Functional Materials MicroNano Devices, Department of Physics, Renmin University of China, Beijing 100872, China. [7]Max-Planck-Institut für Festkörperforschung, Heisenbergstrasse 1, D-70569 Stuttgart, Germany. [8]Collaborative Innovation Center of Advanced Microstructures, Nanjing 210093, China. [9]Shanghai Research Center for Quantum Sciences, Shanghai 201315, China. [10]These authors contributed equally: Boqin Song, Tianping Ying, Xianxin Wu, Wei Xia, Qiangwei Yin. ✉e-mail: ying@iphy.ac.cn; hlei@ruc.edu.cn; guoyf@shanghaitech.edu.cn; shiyan_li@fudan.edu.cn

unconventional superconductivity[19–25], exotic CDW ordering[26–32] and nematicity[33,34]. At ambient conditions superconductivity with a critical temperature of $T_c \sim 0.9$–$2.5$ K emerges inside the CDW order, occurring around $T_{CDW} \sim 78$–$103$ K[16,22,23]. The nature of superconductivity is still controversial due to the conflicting experimental results[21,35,36]. In the parent phase, the CDW ordering exhibits a variety of intriguing properties. Scanning tunneling microscopy (STM) measurements proposed a $2 \times 2$ in-plane[26,37,38] and a $2 \times 2 \times 2$ three-dimensional (3D)[28,39] reconstruction and chiral nature of the CDW ordering[26]. In addition, recent zero-field muon spin relaxation (μSR) measurements revealed an enhanced relaxation rate[40,41] and the optical signatures including the polarization rotation and the magneto-optical Kerr effect were observed[42–44], implying an intrinsic time-reversal symmetry breaking. This can be intimately related to the observed giant anomalous Hall effect[45,46]. Moreover, a nematic ordering emerging from this CDW was suggested in STM[33], magneto-resistance[34], and optical investigations[42,43]. All the evidences tend to suggest a possible loop current ordering[29,43,44] with an electronic origin. However, recent angle-resolved photoemission spectroscopy (ARPES) measurements revealed that the CDW band splitting are consistent with the electronic band structures with an imposed distortion from softening phonon modes[47–49]. The variation of phonon modes across the CDW transition from neutron and Raman scattering indicates their important role in promoting CDW ordering[50–52]. Therefore, combining all the available evidences, electronic interaction and electron-phonon coupling (EPC) may cooperate to generate the CDW.

Both superconductivity and CDW in AV₃Sb₅ can be tuned by external pressure[19,20,53,54], chemical pressure[55], and doping[56]. The kagome metals exhibit superconducting domes under either pressure or doping, but the CDW order is always significantly suppressed by pressure or doping, under which it vanishes abruptly[19,20,53–56]. In thin flakes, the dimension reduction does not only introduce charge modulations but also boosts quantum fluctuations and electronic correlations. Previous investigations of thick flakes reported a competing nature between superconductivity and CDW[57]. However, atomically thin flakes approaching the 2D limit which may exhibit distinct physical properties from the bulk counterpart, remain unexplored due to the challenging exfoliation process.

In this paper, we study CsV₃Sb₅ thin flakes approaching the monolayer limit and trace the collective behavior of the CDW order and superconductivity through direct electrical resistance and Raman measurements. With the reduction in the thickness of the flakes, a dramatic enhancement of the $T_{CDW}$ is observed after a minimum value of 72 K at a critical thickness of around 25 layers, and it reaches a record-high of 120 K in the 5-layer flake, much higher than its bulk value of 94 K. STEM analyses of the exfoliated thin flakes confirm that the kagome crystal structure is preserved with its thickness reduced to at least 4 layers. Raman spectroscopy measurements reveal a weakened electron-phonon coupling in the thin-flake regime, suggesting that a crossover of the mechanisms for the CDW from phonon-driven to dominantly electron-driven. Furthermore, CsV₃Sb₅ can be cleaved even to a monolayer and a metal-insulator transition has been observed for the first time in thin flakes crossing 5 layers. The observed anomalous enhancement of the CDW and its non-monotonic evolution with the reduction of sample thickness reveal fascinating characters of the CDW order, which shines lights on its origin. Our findings establish kagome metals as an ideal system to investigate the competition, coexistence, and collaboration of electron-electron and electron-phonon interactions.

## Results

### Atomic resolution images approaching 2D limit

Figure 1a displays the layered hexagonal lattice of CsV₃Sb₅, consisting of alternately stacked V₃Sb₅ tri-layers and Cs layers. Each V₃Sb₅ tri-layer contains a 2D vanadium kagome net interweaved by a hexagonal lattice of Sb atoms. The semi-metallic nature of antimony endows these kagome metals with moderate A-Sb interlayer coupling, offering a way to exfoliate them into thin flakes. The exfoliation research on tens- or hundreds-layer flakes with ideal CsV₃Sb₅ kagome structure reveals a rapid increase of $T_c$ and quick damping of the $T_{CDW}$ in the vicinity of the bulk limit[57]. Owing to difficulties in controllable exfoliation and surface oxidation affecting the structure stability, few research has been done for thinner flakes and only scattered data points are available so far. Therefore, it is imperative to first address the crucial question of whether the V₃Sb₅ kagome tri-layer structure can survive when the flakes are thinned toward the atomic limit. To answer this, we resort to STEM measurements, which is a powerful tool to identify crystal structures. A piece of CsV₃Sb₅ thin flake is first peeled off, transferred to a Cu grid, and inserted into the STEM chamber while minimizing the exposure to air. Due to the lack of carbon support, we find the edge area will naturally curl (see Supplementary Fig. 1) to expose its transverse section, from which we could determine the edge to be 2 L. Using the intensity contrast of the 2 L sample as a gauge, the sample thickness of the stair structure shown in Fig. 1c is determined to be 4–9 L from the intensity line profile (Supplementary Fig. 2). Inset is the transmission electron microscopy (TEM) image of the 4 L sample acquired from the broken square region. These sharp diffraction spots with an identical diffraction pattern to that of CsV₃Sb₅ suggest that the kagome structure can stabilize down to at least 4 L.

To clearly reveal the loss of surface Cs layers and the stability of the kagome structure in the 2D limit, we further perform a quantitative analysis on an atomic resolution image of the 8 L region, as the zone axis of the bent 4 L region lies far away from the [00l] direction. The Cs, V, and Sb atoms in a kagome lattice can be readily identified in the ABF-STEM image (Fig. 1d) and HAADF-STEM image (Supplementary Fig. 3). As the integrated intensity is proportional to $Z^2$ (Z is the atomic number of the elements), a model of 8 L sample without the outmost Cs layers corresponds best to both the observed HAADF-STEM image (Supplementary Fig. 3c, d) and the intensity line profile (Supplementary Fig. 3e, f). This observation indicates that only the top surface Cs atoms are removed by exfoliation, while the second Cs layer from the surface remains intact and is protected by the outmost V₃Sb₅ layer. therefore, we have a good reason to believe that the sandwiched Cs layers in the 4 L sample or even thinner ones may also be preserved. This is the first and direct observation that the V₃Sb₅ kagome structure can survive at room temperature approaching the 2D limit (at least down to 4 L), and thus lays a solid foundation for our later measurements.

### Anomalous enhancement of the CDW

We are now ready to explore the intrinsic properties of the CDW and superconductivity approaching the 2D limit. All the CsV₃Sb₅ flakes from one monolayer to 155 L measured here are acquired using the latest developed Al₂O₃-assisted exfoliation method[58] (see Supplementary Fig. 4). Gold electrodes are evaporated and the fabricated devices are further encapsulated by BN thin flakes. These protected samples are quite stable and highly reproducible with respect to the thickness (see Supplementary Fig. 5). Figure 2a, b shows the temperature-dependent resistance with a variation of flake thickness. We find that the CDW transition temperature, as indicated by a kink (arrow) in Fig. 2a, gradually decreases from 94 K to 72 K from the bulk sample to 27 L. This evolution is more apparent from the peak positions in Fig. 2b, which displays the derivative of the temperature-dependent resistance. Meanwhile, as shown in Fig. 3c, the onset of $T_c$ smoothly increases from the bulk value of 3.6 K to the highest value of 4.5 K in the 27 L sample. These observations are almost quantitatively consistent with the study on thick flakes[57] and qualitatively consistent with research using other tuning methods, such as external pressure and chemical doping[19,20,53–56]. With further reduction of the sample

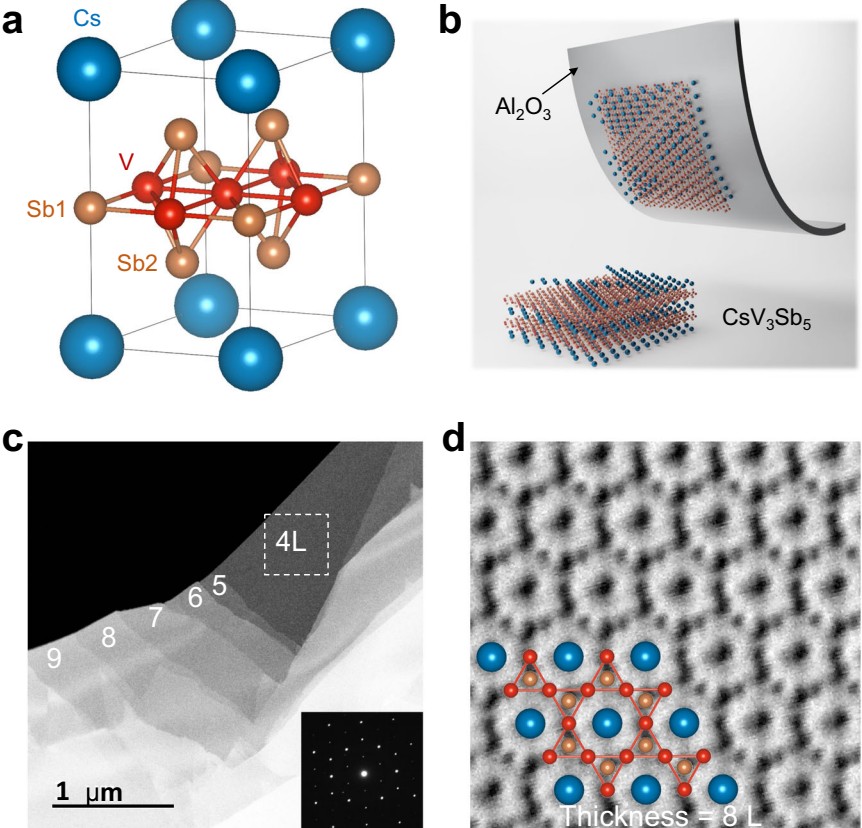

**Fig. 1 | Structure and exfoliation of CsV₃Sb₅ thin flakes. a** Crystal structure of $CsV_3Sb_5$. **b** Illustration of the exfoliation of thin flakes and the loss of surface Cs. The schematic structure of an eight-layer thin flake without surface Cs can be found in Supplementary Fig. 3b. **c** Low-magnification STEM image of the exfoliated flakes. The determination of the thickness can be found in Supplementary Note 2. The selected area electron diffraction (SAED) of the white broken line area (4 layers) is shown in the inset. **d** Annular bright-field (ABF)-STEM image of CsV₃Sb₅ with the thickness of 8 layers. We use 4 L for 4 layers as a shorthand and the same goes for other thicknesses.

thickness, the $T_{CDW}$ surprisingly begins to increase dramatically and surpasses its bulk value of 94 K in the 20 L sample, and finally reaches a record-high temperature of 120 K in the 5 L sample. On the contrary, the $T_c$ decreases rapidly and reaches a minimum value of 0.9 K at 5 L, generating a thickness-dependent superconductivity dome. This non-monotonic evolution and dramatic enhancement of $T_{CDW}$ are in sharp contrast to all previous pressure and doping experiments[19,20,53–56] in these kagome metals, where external tuning knobs always suppress the CDW order and completely quench it over a certain threshold. This implies an intriguing origin of the CDW order in atomically thin flakes, which might be different from the bulk samples.

To explore the mechanism of the CDW order, we first study the anomalous Hall effect (AHE) observed in bulk $CsV_3Sb_5$, which exhibits an intimate relationship with the CDW order[46]. A systematic measurement about thickness dependence of AHE has not been done yet. We performed the Hall measurements at 5 K for various sample thicknesses. The field-dependent $\rho_{xy}$ curves follow a straight line at high temperatures and gradually deviate the linear behavior crossing the $T_{CDW}$ (Supplementary Fig. 6), in line with previous reports. When the sample thickness is reduced from 155 L to 7 L, the slops of the $\rho_{xy}$ curves at high magnetic field remain positive (derived carrier density and mobility are shown in Supplementary Fig. 7) but decline systematically to a smaller value, which can be attributed to the hole doping that caused by the loss of surface Cs atoms. Then, we extract the anomalous Hall resistance $\rho_{xy}^{AHE}$ by subtracting the local linear term[45,46] (Supplementary Fig. 8) and summarize $\rho_{xy}^{AHE}$ as a function of the magnetic field for five samples with distinct thickness in Fig. 2d. Interestingly, with a reduction of thickness, the antisymmetric sideway

"S" line shape first gradually turns flat and then the slope around the origin changes the sign around 27 L and the anomalous Hall resistance gets enhanced rapidly. This indicates that the $\rho_{xy}^{AHE}$ changes its sign from negative in 27 L to positive in 22 L. It can be inferred from Fig. 2d that the AHE vanishes in the samples with thickness about 25 L. Generally, the AHE occurs in the system with time-reversal symmetry (TRS) breaking. When TRS is broken, it opens a gap at the Dirac point near the $M$ point in $CsV_3Sb_5$, resulting in a large Berry phase and giving rise to the intrinsic AHE. In the bulk, the AHE inside the CDW phase is believed to be one of the evidences supporting the TRS breaking of the CDW order, which is further confirmed by the recent $\mu SR$[40] and MOKE[44] measurements. In the thin flakes, we also observed AHE inside the CDW order, suggesting that the CDW order in both bulk and thin flakes breaks TRS. Interestingly, the sign inversion coincides with the upturn of the $T_{CDW}$, confirming the intimate correlation between AHE and CDW in thin flakes. These observations suggest that TRS breaking always occurs in the CDW phase of $CsV_3Sb_5$ and survives even approaching the 2D limit.

### Raman spectroscopy across the critical thickness of 25 L

As canonical CDW ordering generally couples with structural transitions, Raman spectroscopy serves as an ideal technique to study the lattice degree of freedom, which can provide insights into the CDW mechanism. Previous Raman studies in kagome metals have focused on bulk materials or thick flakes[28,50,51]. To explore the thickness-dependent evolution of the CDW order, we focus on the phonon modes evolution of the two regimes: CDW-weakened (above 25 L) and CDW-enhanced (below 25 L). Figure 3a, b shows the representative

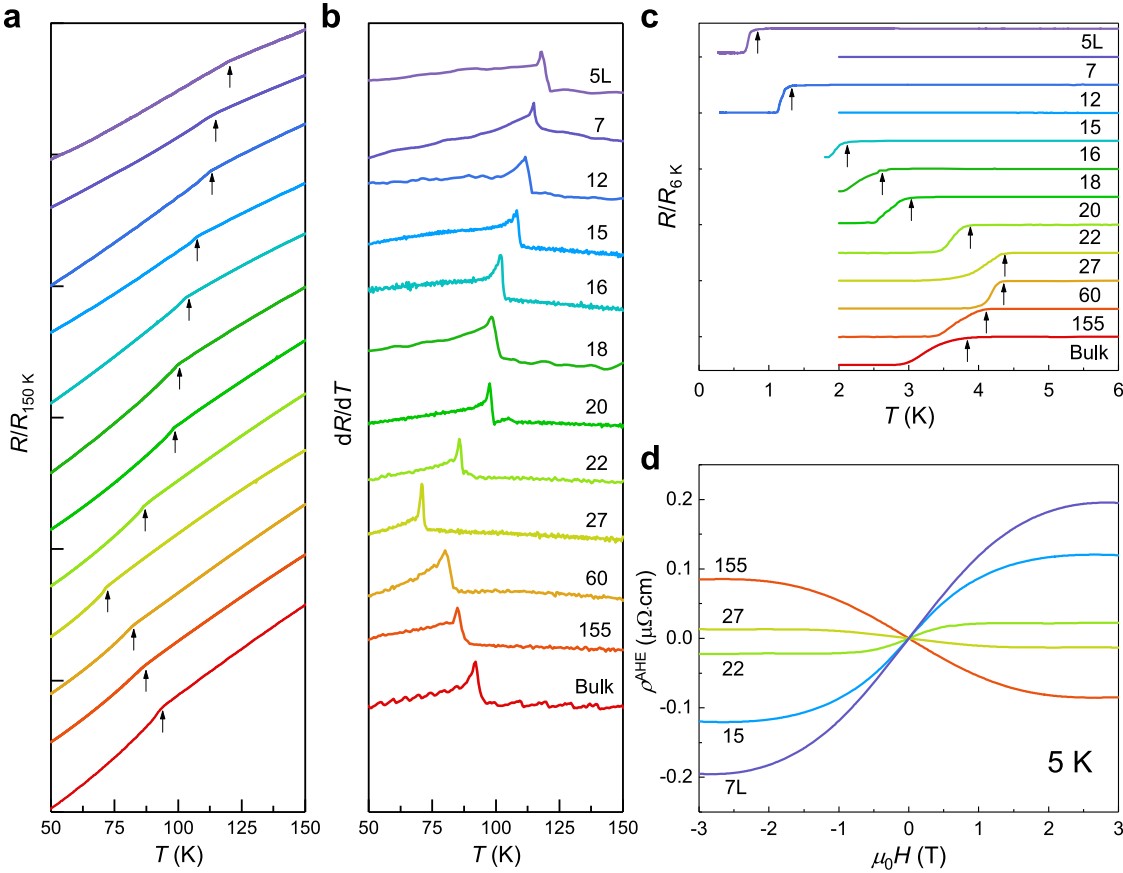

**Fig. 2 | Electrical transport measurements of CsV₃Sb₅ thin flakes. a** Evolution of the CDW kink with a variation of flake thickness. All the data is vertically shifted for clarity. The transition temperatures $T_{CDW}$ are determined from the kink in the $R-T$ curve, which gives consistent results by taking the first derivative $dR/dT$ as shown in (**b**). **c** Thickness-dependent $T_c$ of CsV₃Sb₅. All the curves are vertically shifted. **d** Extracted anomalous Hall effect with the variation of sample thickness. The raw data can be found in Supplementary Fig. 9.

temperature-dependent Raman intensity contour plot for a 45 L and a 15 L area from the same exfoliated flake (The spectra see Supplementary Fig. 9 and Supplementary Fig. 10). The general feature is in line with previous studies in which the absence of softening modes was evidenced by hard-X-ray[28] and neutron scattering[52]. While the folding modes in the bulk can be clearly seen in Supplementary Fig. 11, their intensity decreases rapidly in thin flakes. To visualize the folding modes, we extend the acquisition time to 1 h (3600 s) and scale the spectra to the highest phonon mode ($E_{2g}$) at 117 cm⁻¹. Under such a condition, we successfully observe the CDW-induced folding modes in the thin flakes with thickness down to 37 L (Supplementary Fig. 12), which is consistent with the observed modes in the bulk. Group analyses (Supplementary Note 7) confirm the observed main lattice modes to be $A_{1g}$ and $E_{2g}$, where $A_{1g}$ represents the vibration of Sb2 atoms perpendicular to the *ab*-plane and $E_{2g}$ is the circular vibration mode within the *ab*-plane. The CDW distortion (star-of-David or inverse-star-of-David) mainly involves the motion of V atoms in the kagome layers, and it further couples to an out-of-plane vibration of the Sb2 atoms. Thus, the $A_{1g}$ mode serves as an ideal probe to monitor the coupling between the CDW and the lattice. Figure 3c, f shows the frequency shift of the $A_{1g}$ mode of 45 L and 15 L flakes, respectively. For the 45 L sample, the $A_{1g}$ frequency exhibits an abrupt increase below $T_{CDW}$, as shown in Fig. 3c. The solid black line is the fitting curve using an anharmonic phonon decay (APD) model[59] (Supplementary Note 8) and the dramatic change in slope across the transition of the $A_{1g}$ mode clearly deviates from it, indicating a strong EPC[50,51]. Concomitantly, the corresponding line width and integrated peak intensity sharply decrease below $T_{CDW}$, as shown in Fig. 3d, e, which can be understood

as a reduced EPC derived from the CDW partially gaping out the Fermi surfaces. This phononic renormalization across the $T_{CDW}$ is consistent with earlier results on the bulk sample and provides clear evidence of a significant electron-phonon interaction on the thick flake side[50,51].

Interestingly, the $A_{1g}$ mode in the CDW-enhanced regime (15 L) exhibits a similar hardening and linewidth narrowing but without a sharp increment in frequency crossing the $T_{CDW}$ (Fig. 3f). The whole temperature range of frequency shift and FWHM of our 15 L sample can be well fitted by an APD model[59] (Fig. 3f, g). As demonstrated in Fig. 1, the stable V kagome lattice down to at least 4 L rules out the scenario of crystal degradation in thin flakes, indicating a significant decrease of the EPC with the out-of-plane phonon modes. Note that the intensity of the spectra is dependent on the thickness of the exfoliated samples, however, the temperature-dependent frequency of the $A_{1g}$ mode is related to the CDW order but not the thickness, according to our long-collection-time experiment shown in Supplementary Fig. 13. To qualitatively estimate the strength of the EPC with a variation of flake thickness, we plot $\Delta\omega$ which is defined as the difference between the experimental frequency of the $A_{1g}$ mode and the extrapolated APD fitting at 10 K in Fig. 3i. We find that the difference of $A_{1g}$ mode shift steadily shrinks from 1.2 cm⁻¹ in the bulk to 0.6 cm⁻¹ in the thick flake region and finally vanishes below 25 L. The broken line is a guide to the eye, which ostensibly suggests the disentanglement of the electronic and lattice parts of the CDW order parameter across the critical thickness. The gradual decline of the EPC with decreasing thickness can account for the suppression of CDW order above 25 L, while the vanishing of frequency shift of the $A_{1g}$ mode below 25 L suggests a non-phonon-driven enhancement of the CDW order.

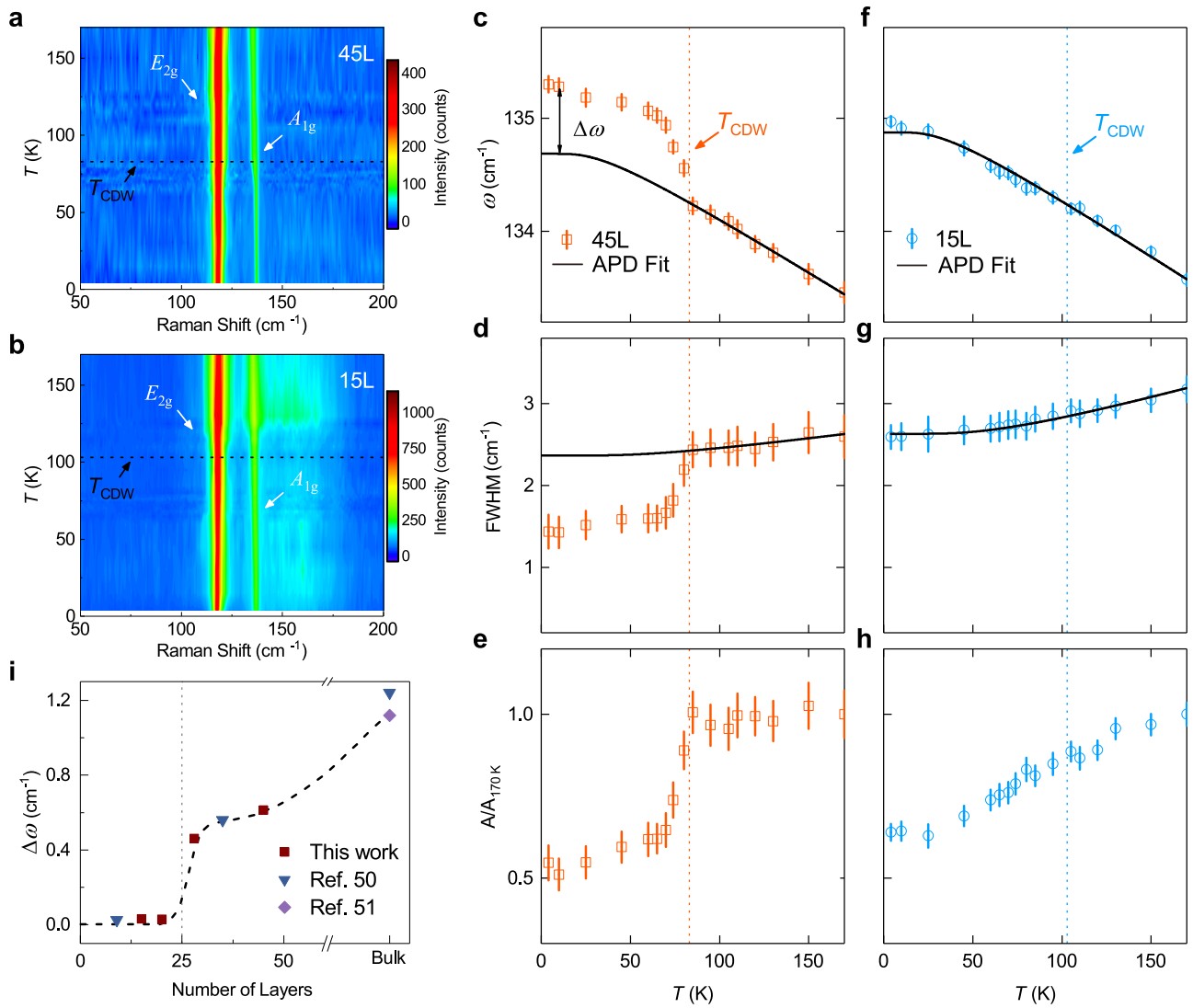

**Fig. 3 | Thickness-dependent Raman response of $CsV_3Sb_5$ measured in the XX configuration. a, b** Color contours of $E_{2g}$ and $A_{1g}$ modes with the evolution of temperature for $CsV_3Sb_5$ thin flakes with the thickness of 45 L and 15 L, respectively. **c–h** Fitted frequency ($\omega$), linewidth (FWHM), and normalized integrated peak intensity ($A/A_{170\,K}$) of the $A_{1g}$ main lattice phonon for the 45 L (**c–e**) and 15 L (**f–h**) samples. Error bars are determined from fitting. Black lines are fitted by APD model. **i** Difference between the $A_{1g}$ frequency ($\omega_{exp}$) and the APD fitted value ($\omega_{fit}$) at 10 K with the variation of sample thickness. $\Delta\omega$ represents $\omega_{exp} - \omega_{fit}$ shown in (**c**). The broken line is a guide for the eye that highlights the abrupt vanishing of the difference across the critical thickness of 25 L.

## Metal-insulator transition approaching 2D limit

To give a complete picture of $CsV_3Sb_5$ covering the whole thickness range, we exfoliate and measure the thin flakes down to the monolayer limit. The optical image of 1L-5L samples is shown in the lower inset of Fig. 4a. Despite the 5 L sample showing a CDW kink at 120 K and a superconducting transition at 0.9 K (Fig. 2), in the sample with one layer less (4 L), no kink feature can be identified (Supplementary Fig. 14), and superconducting transition cannot be detected down to 0.3 K. Since the kagome framework of $V_3Sb_5$ is preserved in the 4 L sample, the absence of CDW kink and superconductivity may be ascribed to the charge doping from the loss of surface Cs by exfoliation. From 5 L to 2 L, a thickness-dependent metal-insulator transition can be observed with a negative slope of the $R-T$ curve in the 3 L sample over the whole temperature range. The resistance of the monolayer with the complete loss of Cs turns out to be a good insulator. As shown in the upper inset of Fig. 4a, the resistance of the 2 L sample follows a 2D variable-range-hopping behavior above 4 K[60]. With further decreasing the temperature, a clear deviation from linear behavior can be seen. In the lower temperature region (below 3 K), the resistance can be fitted in the framework of opening a Coulomb gap

(Efros-Shklovskii gap, see Supplementary Fig. 15)[61], indicating a sizable electron-electron interaction in thin flakes with less than 4 L, although some other possibilities, such as crystalline imperfections approaching the atomic limit, buckling due to 2D fluctuations, and percolation through grain boundaries, cannot be excluded. The unusual insulating behavior accompanied by an absence of CDW kink approaching the 2D limit is reported for the first time in kagome metals.

In Supplementary Fig. 16, the impact of the evaporated $Al_2O_3$ layer on the physical properties of the exfoliated thin flakes was studied and it was found that the $Al_2O_3$-assisted exfoliation process does not affect the $T_{CDW}$, as well as the overall transport behavior. Therefore, the observed insulating behavior may be related to the heavy hole doping and enhanced correlation effects. As the CDW can partially gap out the Fermi surfaces, this insulating phase may be a Mott or other correlated phases, which offers an interesting parent phase for future gating experiments. The present study demonstrates unique characteristics of the CDW order in kagome metals approaching the 2D limit and may stimulate further investigations on its mechanism and interplay with superconductivity in thin flakes with diverse tunable approaches.

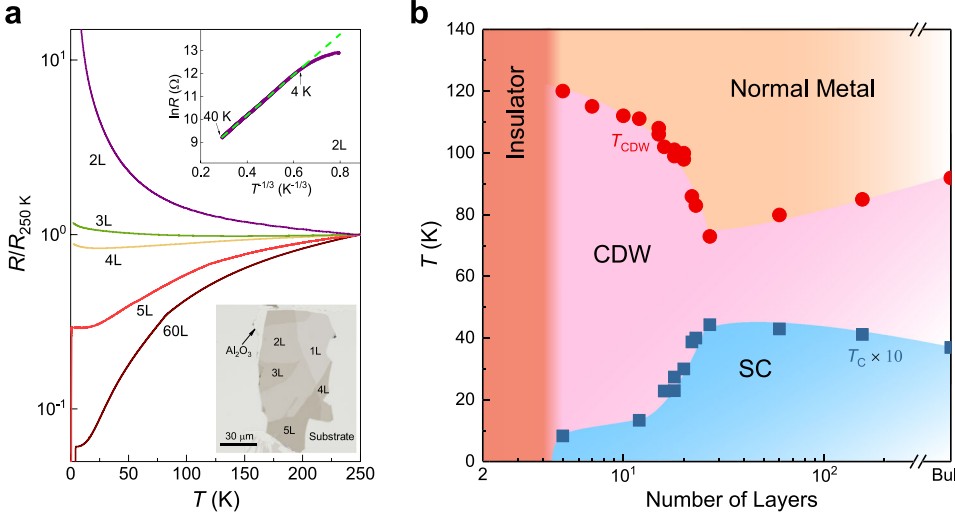

**Fig. 4 | Thickness-dependent resistance and phase diagram of CsV$_3$Sb$_5$ approaching the 2D limit. a** Temperature-dependent resistance of CsV$_3$Sb$_5$ with different flake thicknesses from 2–5 layers. A sample with 60 L is included for comparison. Monolayer CsV$_3$Sb$_5$ turns out to be an insulator and its resistance cannot be measured. All the data are normalized at 250 K. The upper inset shows the 2D variable-range-hopping behavior of the 2 L sample. The lower inset is the optical image of 1–5 layers exfoliated on an Al$_2$O$_3$ substrate. **b** Phase diagram of CsV$_3$Sb$_5$ with the variation of sample thickness.

## Discussion

We summarize our observations of CsV$_3$Sb$_5$ kagome flakes in the temperature-thickness phase diagram, shown in Fig. 4b. The $T_{CDW}$ gradually decreases with decreasing thickness from the bulk limit and reaches the minimum at ~25 L but increases abruptly with further reducing thickness. In contrast, the $T_c$ exhibits an opposite behavior and forms a superconducting dome. For flakes with less than 5 L, an exotic MIT transition occurs. The non-monotonic evolution of $T_{CDW}$ with a reduction of flake thickness is rare. In the bulk AV$_3$Sb$_5$, the CDW order exhibits a 3D character with $2 \times 2 \times 2$ or $2 \times 2 \times 4$ structural reconstruction[28,39,62,63]. It is generally believed that EPC plays a crucial role in promoting the CDW order[47,50–52] in bulk kagome metals despite the absence of phonon softening across the transition, which may be related to the failure of the conduction electrons to screen the phonon vibration[50,64]. The 3D CDW involving in-plane V distortion can naturally couple with the out-of-plane Sb2 vibration, i.e., the $A_{1g}$ mode[52,62]. The gradual suppression of the CDW order with decreasing thickness above 25 L can be attributed to the weakening of EPC, indicated by the decreasing $A_{1g}$ frequency shift (Fig. 3c). An intuitive understanding is that the exfoliation will reduce the interlayer coupling and consequently suppress the 3D CDW ordering along the c axis and its coupling to the $A_{1g}$ mode, thus a 2D CDW[26,37,38], in line with recent theoretical prediction[65] and experimental studies[66,67]. Moreover, the frequency shift vanishes below 25 L, indicating the decoupling between the CDW order and $A_{1g}$ mode which induces in-plane CDW distortion by coupling to the motion of V atoms in the EPC scenario. Thus, the observed dramatic enhancement of $T_{CDW}$ (Fig. 4b) should have a non-phonon origin.

Dimension reduction imposed thermal fluctuations will generally overwhelm the quantum confinement, leading to a decline of the transition temperature in canonical CDW materials[68–70] with few exceptions[71,72]. To the best of our knowledge, the enhancement of $T_{CDW}$ has only been reported in 2H-NbSe$_2$ and VSe$_2$ thin flakes without signatures from electrical transport measurements and are now believed to be related to the strengthening of EPC[71,72]. The dramatic enhancement of $T_{CDW}$ and the vanishing $\Delta\omega$ across the critical thickness makes CsV$_3$Sb$_5$ distinct from all known materials. Thickness reduction in CsV$_3$Sb$_5$ can enhance the electronic interaction and the VHSs in the vicinity of the Fermi level become more relevant as they contribute to divergent DOS in 2D[73–75]. Therefore, the dramatic enhancement of $T_{CDW}$ might be attributed

to enhanced correlation effects. In this scenario, the CDW order is $2 \times 2$ and dominantly electronically driven with less atomic distortion in the V kagome lattice. The Sb2 out-of-plane distortion in this 2D CDW should be greatly reduced. Thus, it suppresses the kink of Sb2 $A_{1g}$ mode with out-of-plane vibration across the CDW transition.

The power-law fitting of the low-temperature normal state resistance gives almost comparable parameters $\alpha$ across the entire range of sample thicknesses (Supplementary Fig. 17). Interestingly, the linear fitting of the high-temperature resistance before entering the CDW order shows a sharp drop of the slope crossing the critical thickness of 25 L, as shown in Supplementary Fig. 18, indicating its close relationship with the anomalous enhanced $T_{CDW}$. Upon detailed analysis of the superconducting transition width, as shown in Supplementary Fig. 19, we observe two interesting features that are worth mentioning. The first is a gradual decline of the transition width with thinner samples, attributed to decreased inhomogeneity. The second is a prominent jump around the critical thickness of 25 L, which warrants further investigations. A plausible explanation is the existence of competing orders or a crossover between different electronic states in the vicinity of 25 L.

The persistence of the AHE down to at least 7 L indicates the survival of the CDW-related time-reversal symmetry breaking under the crossover from EPC-driven CDW to dominantly electronically driven CDW. Therefore, the electronic structure of CsV$_3$Sb$_5$ kagome lattice may play a crucial role in the origin of time-reversal symmetry breaking. The sign reversal of the anomalous Hall resistance may derive from the hole-doping-induced sign change in the integral of Berry curvature[29,76]. Superconductivity is weakened with the enhancement of CDW order due to their competing nature. Collectively, the crossover from EPC to electronic interaction can provide a reasonable explanation for our experimental observations in CsV$_3$Sb$_5$ flakes. It will be very interesting to study the nature of the CDW order in the enhanced regime, including dimensionality, electronic structure, and symmetry breaking, which may be distinct from the bulk. Further investigation such as nano ARPES measurements and micro magneto-optical Kerr effect might provide informative knowledge in this regard.

We systematically investigate the crystal structure, electrical transport, and Raman spectroscopy of CsV$_3$Sb$_5$ approaching its monolayer limit. The kagome structure can be preserved to at least 4 L,

which lays a solid foundation for a consistent discussion of the transport and Raman measurements. We discovered two critical thicknesses in the thin flake region. The most prominent one at 25 L is where the dramatic enhancement of the CDW starts, where the superconducting $T_c$ is optimal and where the sign reversal of the AHE occurs. A mixed origin of phonon-driven and electron correlation is suggested to be the driving force of CDW, while the sample thickness serves as a tuning knob to alter the dominant role. The other critical thickness at 4 L signifies the onset of MIT, where correlations and fluctuations come into play. Unconventional phenomena can be anticipated by electrical gating of CsV$_3$Sb$_5$ at these critical thicknesses (25 L, 4 L) and for monolayer. The present results also impose crucial constraints on the theoretical models for understanding the fascinating CDW order, superconductivity, van Hove singularities, as well as their mutual interdependence.

## Methods

### Crystal synthesis
The CsV$_3$Sb$_5$ bulk crystals used in this study were grown from Cs ingot (purity 99.9%), V powder (purity 99.9%), and Sb grains (purity 99.999%) using the self-flux method. The mixture was put into an alumina crucible and sealed in a quartz ampoule under partial argon atmosphere. The sealed quartz ampoule was heated to 1273 K for 12 h and soaked there for 24 h. Then it was cooled down to 1173 K at a rate of 50 K/h and further to 923 K at a slower rate. Finally, the ampoule was taken out of the furnace and decanted with a centrifuge to separate CsV$_3$Sb$_5$ single crystals from the flux.

### TEM measurements
The CsV$_3$Sb$_5$ thin flake was transferred to a Cu grid, and inserted into the STEM chamber with minimum exposure to air. Detailed characterization of the transverse section area was conducted using high-resolution transmission electron microscopy (HRTEM, Tecnai F20 super-twin). Atomically resolved intensity for quantitively analysis was obtained using high-angle annular dark-field scanning transmission electron microscopy (HAADF-STEM, JEOL ARM200F).

### Devices fabrication and Electrical transport measurement
We obtain few-layer CsV$_3$Sb$_5$ samples using an Al$_2$O$_3$-assisted exfoliation technique[58]. First, an Al$_2$O$_3$ thin film is thermally evaporated onto a freshly prepared surface of the bulk crystal. Then, we lift the Al$_2$O$_3$ thin film using a thermal release tape, accompanied by CsV$_3$Sb$_5$ thin flakes cleaved from the bulk crystal. Subsequently, the Al$_2$O$_3$/CsV$_3$Sb$_5$ stack is released onto a piece of polydimethylsiloxane (PDMS) with the CsV$_3$Sb$_5$ side in contact with the PDMS surface. The thin flakes are finally stamped assembly onto a sapphire substrate and inspected under an optical microscope in transmission mode. All the devices are fabricated in an argon atmosphere with O$_2$ and H$_2$O content kept below 0.5 parts per million to avoid sample degradation. The thickness of CsV$_3$Sb$_5$ thin flakes was determined by optical contrast after being calibrated by an atomic force microscope (AFM) (Park NX10). A Quantum Design physical property measurement system (PPMS) and a $^3$He refrigerator are used for transport measurement with Stanford Research 830. Resistance of samples is measured using the four-probe method, while Hall measurements are taken in a five-wire configuration. A rotator insert is used to tilt the angle between the magnetic field and the $c$ axis.

### Raman measurements
The freshly exfoliated CsV$_3$Sb$_5$ flakes were transferred to a silicon substrate coated with 300 nm SiO$_2$, and capsulated by BN thin flakes to eliminate the possibility of air exposure. Raman scattering experiments were performed in a back-scattering geometry with the excitation line $\lambda = 532$ nm of a solid-state laser. The laser beam was focused on the sample at a micron-sized spot. The spectra were recorded using a WITec $\alpha$-300R and JY Horiba HR800 spectrometer. The temperature was controlled by a HFS600E LinKam stage.

## Data availability
The data that support the findings of this study are available from the corresponding authors upon reasonable request.

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

## Acknowledgements

We thank Dr. Luojun Du and Dr. Kun Jiang for fruitful discussions. We gratefully acknowledge Dr. Yang Yang and Dr. Luojun Du for help with the low-temperature Raman data acquisition and analysis. This work is supported by the Natural Science Foundation of China (Grant No. 12174064) and the Shanghai Municipal Science and Technology Major Project (Grant No. 2019SHZDZX01). Y.F.G. was supported by the Major Research Plan of the National Natural Science Foundation of China (No. 92065201) and the Program for Professor of Special Appointment (Shanghai Eastern Scholar). H.C.L. was supported by Beijing Natural Science Foundation (Grant No. Z200005), the Ministry of Science and Technology of China (Grant No. 2018YFE0202600), and National Natural Science Foundation of China (Grant No. 12274459). T.P.Y. was supported by the National Key Research and Development Program of China (No. 2021YFA1401800), and the Natural Science Foundation of China (Grant Nos. 52272267, 52250308).

## Author contributions

S.Y.L. and T.P.Y. conceived the idea and designed the experiments. B.Q.S. and X.F.Y. performed the electrical transport measurements. W.X., Q.W.Y., H.C.L., and Y.F.G. synthesized the single crystal samples. Q.H.Z., T.P.Y., and L.G. performed STEM measurements. B.Q.S., Y.P. S., and J.G.G. performed the Raman measurements. X.X.W., A.P.S., J.P.H., and X.L.C. performed the theoretical analyses. B.Q.S., T.P.Y., X.X.W., and S.Y.L. analyzed the experimental data and wrote this paper with comments from all authors. B.Q.S., T.P.Y., X.X.W., W.X., and Q.W.Y. contributed equally to this work.

## Competing interests

The authors declare no competing interests.
