## [Peer Review File · Nature Communications]

Anomalous enhancement of charge density wave in kagome superconductor CsV₃Sb₅ approaching the 2D limitREVIEWER COMMENTS

Reviewer #1 (Remarks to the Author):

The manuscript by Boqin Song, et al describes a systematic layer-dependent study on charge density wave (CDW) and superconducting state (SC) in a recently discovered Kagome material CsV₃Sb₅. They found 1) A critical thickness at 25 layers where the transition temperature of CDW is the lowest, and T_c of SC state the highest, and a sign reversal of AHE happens, suggesting a competing order of CDW and SC. They suggest a crossover from electron-phonon coupling to electronic interaction-driven CDW by comparing temperature-dependent anharmonic phonon decay with the CDW formation temperature. 2) Another critical thickness at four layers where a metal-insulator transition may happen.

This is an in-time and rigorous work that could stimulate works exploring those Kagome compounds in the 2D limit, where intriguing physics (such as quantum anomalous hall effect, gating effect, etc.) could happen. It may warrant its publication in Nature communications after addressing the following concerns:

1. The authors adopted Al₂O₃-assisted exfoliation and stencil mask to fabricate the devices. This fabrication process allowed them to electrically probe CDW, SC, and anomalous Hall effect (AHE) down to the bilayer and protect the sample from degradation. To validate that the Kagome structure survives, structural information was obtained by STEM down to four layers. There is no clear degradation in Supplementary Fig.3. However, the bottom-most few layers may degrade when evaporating Al₂O₃; thus, the <4 layers flakes show insulating behaviors. Can the authors comment on this possibility? Did the authors try to use the same or similar Al₂O₃-assisted exfoliation to prepare a sample for STEM for the control experiment?
2. Because loss of surface Cs tends to hole dope the sample, calculated carrier density from Hall data v.s. thickness could provide vital information for stimulating future work. The mobility can then be calculated.
3. In fig.2c, the SC transition sharpness has a trend to be enhanced in thinner samples, which may be related to the decreased inhomogeneity in thinner samples. Can the author try to plot scaled data near the SC transition to reveal it?
4. The suppression of sharp increment of A_{1g} mode in a thin flake is interesting. However, A_{1g} mode is an indirect probe of CDW because CDW first couples to V atoms and then couples to the out-of-plane vibration of Sb₂ atoms. The sensitivity of A_{1g} mode on the CDW phase may heavily rely on the finite size effect on z direction. A detailed temperature-dependent Raman may give more information. However, this measurement could be experimentally heavy. The authors must admit all the possible explanations for this specific phenomenon without this experiment.
5. What is the exact resistance of this 2L sample at low temperature? A μm -scale device with such a large (>G Ω) cannot avoid the contact heating effect. The E-S VRH model fitting in Supplementary Note 7 is thus unconvincing if this is the case.

Minor suggestions:

1. The author should add the formula used for the anharmonic phonon decay model fitting.
2. Supplementary Fig.3b has more important information than the current Fig.1a, such as the loss of surface Cs. The authors may consider revising Fig.1 to add that information.
3. I cannot see the Raman data from Ref.57. There is also more data with different thicknesses in Ref.58's supplement.
4. In Fig.3b, what are those modes near 150cm⁻¹?
5. In Fig.4, dR/dT in 2-4L v.s. 5L may help to reveal the possible absence of CDW clearly.

Reviewer #2 (Remarks to the Author):

In this work, the authors report discovery of an unusual enhancement of charge density wave phase in this CsV₃Sb₅. They observed that the transition temperature to the CDW phase has a minimum for 27 layers and whereas the superconducting transition temperature has a maximum at that thickness. They theorize that this is due to a crossover from electron-phonon coupling (EPC) to dominantly electronic interactions for the CDW phase as the thickness is reduced, but no

theoretical model is offered to support this interpretation. This manuscript contains some very interesting data, although the analysis is somewhat incomplete.

1. In page 7, the authors describe the temperature dependence of the field-dependent ρ_{xy} curves, but I cannot find such data in the manuscript of the SI. Please include the actual experimental data.

2. In page 7, the authors claim "These observations suggest that the intrinsic property of the CDW order, i.e., time-reversal symmetry breaking, always occurs in thin flakes and survives even approaching the 2D limit." How is it different from the bulk case, and why?

3. Although the Raman measurements are the most important part of the work, no Raman spectrum is included. The Raman spectra for different thicknesses and their temperature dependence should be presented explicitly. Without them, one cannot judge the validity of the interpretation. The Raman intensity maps of Fig. 3a and b are not enough.

4. I wonder why only 2 peaks are observed in the Raman spectra. Because the number of atoms per unit cell is very large, I would expect many phonon modes. The authors should include group theoretical analysis of phonon modes and show how many phonon modes are Raman active.

5. The authors claim that the dramatic increase of the CDW transition temperature in thin samples should have a non-phonon origin. Then what is the origin? Can one offer some reasonable explanation?

6. The authors used 2 mW of the laser power focused to a micron-sized spot. This corresponds to a very high intensity, which may well damage the sample. Is there any data that shows that the laser did not cause damage to the specimen?

7. In page 7, the authors refer to Fig. 3d, which should be Fig. 2d.

Reviewer #3 (Remarks to the Author):

Report on the manuscript entitled "Anomalous enhancement of charge density wave in kagome superconductor CsV₃Sb₅ approaching the 2D limit"

The manuscript reports the study of the Cs-based Kagome superconductors family as a function of number of layers down to 2 layers. The authors employed transport and Raman measurements, as well as characterizations such as TEM and X-ray diffraction. This work represents the first comprehensive study on this family in few layers limit and on clean samples. An interesting phase diagram is revealed, with competing CDW and superconductivity but non monotonic behaviour as a function of number of layers. The CDW tends to disappear smoothly down to 20-25L then is strongly reinforced below this value. Below 4L, the system becomes insulator and of course is no more superconductor. The authors claimed that this phase diagram can be accounted by a change of dominant mechanism in the CDW drivers, from EPC to electronically driven.

This study is new and well accomplished, it thus deserves publication as soon as possible. Nevertheless I have some remarks.

First, the Raman results are limited to the two high energy phonons that are already present in the normal phase. The previous Raman studies on bulk system have shown clear new peaks and amplitudons in the CDW. The authors must look for these features, and comment on them. If their apparatus allows it, they can also look for low energy excitations like quasi-elastic line and mode below 50 cm^{-1} . If they have no access to this, many modes are also expected above 70 cm^{-1} . I then strongly encourage to show these results or to perform more Raman experiments down to 2L on the present samples. I also encourage to show the raw spectra in the Supplementary in addition to the color plots.

Secondly, I would be interested to see the transport data without shift. Indeed, the comparison of the Temperature tendency, power-law behaviour ... as a function of number of layers might help to conclude on CDW gap or other dimension-limited behaviour. More information can be extracted from the present transport studies.

Here are additional comments:

P7: the link between TRSB and CDW (through AHE) should be discussed more and clarified.

P9: the authors said: "The gradual decline of the EPC with decreasing thickness can account for the suppression of CDW order at above 25L". Please clarify what you mean ("at above").

P9: the authors said: "the absence of CDW kink and superconductivity may be ascribed to the loss of surface Cs by exfoliation." What do you mean? Do you think this is an effect of change in doping level? Do you know the doping expected from the Cs layer?

P9: The author fitted the resistivity at low temperature. The conclusion for the presence of a coulomb gap may be softened as the fit concerns only 1K range. The fitting alone doesn't prove the existence of Coulomb gap (a gap yes but the mechanism is not proven).

Could the authors clarify the difference between figure S3 c and d?

The authors mentioned that Fig S6 presents some dependence in the main but there is no temperature dependence in Fig. S6.

To conclude, this work is very interesting, new and will be of interest for a large community in condensed matter. At present, the analysis and maybe Raman measurements were not used as much as the authors could have done. More information can be extracted from this study. I then strongly encourage to improve this before publication in Nature Communication.

Response to Reviewers

We thank all the three reviewers for their positive evaluation and for praising the high significance of our work. We thank Reviewer #1 for his/her statements, “This is an in-time and rigorous work that could stimulate works exploring those Kagome compounds in the 2D limit, where intriguing physics (such as quantum anomalous hall effect, gating effect, etc.) could happen. It may warrant its publication in Nature communications...”. We thank Reviewer #2 for pointing out “This manuscript contains some very interesting data”. We also appreciate the positive remarks by Reviewer #3, who states that “This study is new and well accomplished, it thus deserves publication as soon as possible”.

Following the comments and suggestions of all three reviewers, we have revised the manuscript. Their main concern is about the Raman measurements and its interpretation. To address this, we carried out additional detailed experimental studies and a thorough analysis of the Raman spectroscopy, as well as additional transport measurements. We added two new data points (28L and 20L) to show that the frequency change of the A_{1g} mode across the CDW decreases with decreasing thickness. These further strengthen our observation and the proposed scenario. Below is a point-to-point reply to the comments voiced by the reviewers.

Reviewer #1:

The manuscript by Boqin Song, *et al.* describes a systematic layer-dependent study on charge density wave (CDW) and superconducting state (SC) in a recently discovered Kagome material CsV_3Sb_5 . They found 1) A critical thickness at 25 layers where the transition temperature of CDW is the lowest, and T_c of SC state the highest, and a sign reversal of AHE happens, suggesting a competing order of CDW and SC. They suggest a crossover from electron-phonon coupling to electronic interaction-driven CDW by comparing temperature-dependent anharmonic phonon decay with the CDW formation temperature. 2) Another critical thickness at four layers where a metal-insulator transition may happen.

This is an in-time and rigorous work that could stimulate works exploring those Kagome compounds in the 2D limit, where intriguing physics (such as quantum anomalous Hall effect, gating effect, etc.) could happen. It may warrant its publication in Nature communications after addressing the following concerns:

Reply: We thank Reviewer #1 for carefully reviewing of our paper and we appreciate his/her positive and constructive comments for improving our paper. In the revised manuscript, we have performed additional measurements and added more discussions to address the reviewer’s concerns. Below is our response to the reviewer's technical comments.

1. The authors adopted Al_2O_3 -assisted exfoliation and stencil mask to fabricate the devices. This fabrication process allowed them to electrically probe CDW, SC, and anomalous Hall effect (AHE) down to the bilayer and protect the sample from degradation. To validate that the Kagome structure survives, structural information was obtained by STEM down to four layers. There is no clear degradation in Supplementary Fig. 3. However, the bottom-most few layers may degrade when

evaporating Al₂O₃; thus, the <4 layers flakes show insulating behaviors. Can the authors comment on this possibility? Did the authors try to use the same or similar Al₂O₃-assisted exfoliation to prepare a sample for STEM for the control experiment?

Reply: We thank the reviewer for the valuable comment. Indeed, the Al₂O₃-exfoliated few-layer samples are technically not suitable for STEM measurements due to the thick evaporated amorphous Al₂O₃ layer (60~100 nm), which masks the signal of the atomic-thick flakes. As an alternative, we propose a strategy to detect the possible influence of the evaporated Al₂O₃ on the samples. We first prepare a 16L sample with a T_{CDW} of 101.5 K using the Al₂O₃ method (Fig. R1a). After measuring its resistance, we further evaporate another layer of Al₂O₃ on the top of the device and measure its resistance again (Fig. R1b). If the top few layers, say 4 layers, are oxidized, the sample should behave like an 12L sample and the T_{CDW} will be enhanced to 110 K. However, as shown in Fig. R1c, d, the T_{CDW} remains almost invariant, indicating that the Kagome layers beneath the surface are less influenced by the Al₂O₃ deposition. Therefore, we believe that the observed insulating behavior approaching the atomic limit should be intrinsic, rather than derived from the degradation. To clarify this point, we have added a new paragraph in the discussion and incorporated Fig. R1 into the revised Supplementary Information as Supplementary Fig. 16.

To clarify this issue, we added a new paragraph in the Supplementary Information, which reads:

“To determine if the MIT observed below 4L is an intrinsic property or a result of crystalline degradation during the Al₂O₃-assisted exfoliation process, we prepare a 16L sample with a T_{CDW} of 101.5 K using the Al₂O₃ method (Supplementary Fig. 16a). After measuring its resistance, we further evaporate another layer of Al₂O₃ on the top of the device and measure its resistance again (Supplementary Fig. 16b). If the top few layers, say 4 layers, are degraded, the sample should behave like an 12L sample and the T_{CDW} will be enhanced to 110 K. However, as shown in Supplementary Fig. 16c, d, the T_{CDW} remains almost invariant, indicating that the Kagome layers beneath the surface are less influenced by the Al₂O₃ deposition. Therefore, we believe that the observed insulating behavior approaching the atomic limit is intrinsic, rather than derived from the degradation.”

We have also added a sentence in the main text:

“In Supplementary Figure 16, the impact of the evaporated Al₂O₃ layer on the physical properties of the exfoliated thin flakes was studied and it was found that the Al₂O₃-assisted exfoliation process does not affect the T_{CDW} and the overall transport behavior.”

Figure R1. **a, b** A 16L CsV_3Sb_5 sample before and after top- Al_2O_3 deposition. **c, d** Resistance and first derivative dR/dT of the sample before (blue) and after (red) top- Al_2O_3 deposition with their T_{CDWS} at 101.5 K and 101.6 K, respectively.

2. Because loss of surface Cs tends to hole dope the sample, calculated carrier density from Hall data v.s. thickness could provide vital information for stimulating future work. The mobility can then be calculated.

Figure R2. Carrier density and mobility of CsV_3Sb_5 samples with various thickness extracted from high-field Hall data at 4 K. Hole density increase with the reduction of thickness.

Reply: We thank the reviewer for the good suggestion. As shown in Fig. R2, the calculated hole carrier density at 4 K increases as the sample thickness decreases. This trend is particularly pronounced in the thin limit, providing strong evidence for the hole doping effect caused by the loss

of surface Cs atoms. The calculated carrier mobility is shown in Fig. R2. We have incorporated this figure into the revised Supplementary Information.

3. In Fig. 2c, the SC transition sharpness has a trend to be enhanced in thinner samples, which may be related to the decreased inhomogeneity in thinner samples. Can the author try to plot scaled data near the SC transition to reveal it?

Reply: We greatly appreciate the reviewer’s sharp observation and the excellent suggestion. After extracting the SC transition width from Fig. R3a (see Fig. R3b), we observe two interesting features that are worth mentioning. The first is a gradually enhanced SC transition sharpness in thinner samples, which is attributed to the decreased inhomogeneity (as mentioned by the reviewer). The second is a prominent jump at the critical thickness of 25L. Here the origin is still unclear. However, similar behavior has been observed in other materials, such as pressurized CeCoIn_5 [J. Phys.: Condens. Matter **16**, 8905 (2004)] and RbV_3Sb_5 [Phys. Rev. Research **3**, 043018 (2021)], where the highest transition temperature (T_c) and broadening transition width are seen when approaching a pressure quantum-critical point (QCP). These may suggest the existence of competing orders or a crossover between different electronic states in the vicinity of 25L.

We have incorporated the thickness-dependent SC transition width to the Supplementary Information, and added a paragraph in the discussion of the main text:

“The power-law fitting of the low-temperature normal state resistance gives almost comparable parameters α across the entire range of sample thicknesses (Supplementary Fig. 17). Interestingly, the linear fitting of the high-temperature resistance before entering the CDW order shows a sharp drop of the slope crossing the critical thickness of 25L, as shown in Supplementary Fig. 18, indicating its close relationship with the anomalous enhanced T_{CDW} . Upon detailed analysis of the superconducting transition width, as shown in Supplementary Fig. 19, we observe two interesting features that are worth mentioning. The first is a gradual decline of the transition width with thinner samples, attributed to decreased inhomogeneity. The second is a prominent jump around the critical thickness of 25L, which warrants further investigations. A plausible explanation is the existence of competing orders or a crossover between different electronic states in the vicinity of 25L.”

Figure R3. a, b Scaled low-temperature resistance and extracted superconducting transition width as a function of sample thickness.

4. The suppression of sharp increment of A_{1g} mode in a thin flake is interesting. However, A_{1g} mode is an indirect probe of CDW because CDW first couples to V atoms and then couples to the out-of-plane vibration of Sb2 atoms. The sensitivity of A_{1g} mode on the CDW phase may heavily rely on the finite size effect on z direction. A detailed temperature-dependent Raman may give more information. However, this measurement could be experimentally heavy. The authors must admit all the possible explanations for this specific phenomenon without this experiment.

Reply: We thank the reviewer for the pertinent comment and constructive suggestion. We fully agree with Reviewer #1 on that the detection of the A_{1g} mode is an indirect probe of the CDW. The intensity of the spectra is dependent on the thickness of the exfoliated samples, however, the temperature-dependent frequency of the A_{1g} mode is related to the CDW order but not the thickness, according to our measurements. To further support this, we conducted a long-collection-time experiment for two flakes, 28L and 20L, on two sides of the critical thickness of 25L with comparable peak intensity, as shown in Figs. R4 and R5. The results show a clear frequency increment of the A_{1g} mode in the 28L sample across the CDW transition, but not in the 20L one. The frequency change across the CDW transition as a function of flake thickness is plotted in Fig. R5, indicating a weakened electron-phonon coupling (EPC) with decreasing thickness. Further combined with the enhanced T_{CDW} below 25L in experiments, the crossover from EPC to electronic interactions provides a reasonable explanation, although we cannot rule out other scenarios. We have included the two new data points in the Fig. 3i of the revised manuscript.

Figure R4. Raman response of 28L and 20L samples measured in the XX configuration. **a, b** Raman spectra of 28L and 20L collected at 4 K with comparable peak intensities. **c, d** Temperature-dependent frequency of A_{1g} mode for 28L and 20L samples.

Figure R5. Difference between experimental and fitted frequency of A_{1g} . New temperature-dependent experiments on 28L and 20L samples, indicated by points encircled in red, have been updated in this revision.

5. What is the exact resistance of this 2L sample at low temperature? A um-scale device with such a large ($>10^9\Omega$) cannot avoid the contact heating effect. The E-S VRH model fitting in Supplementary Note 7 is thus unconvincing if this is the case.

Figure R6. Raw data of 2L sample from 2-250 K. We have included it in the Supplementary Information.

Reply: We thank the reviewer for the comment. Figure R6 presents the raw data of the 2L sample with a maximum resistance of 0.4 MOhm at 2 K. We have changed the excitation currents (1 μ A, 20 μ A) at 2 K and found that the resistance remained almost constant, indicating that the contact heating effect is negligible.

Minor suggestions:

1. The author should add the formula used for the anharmonic phonon decay model fitting.

Reply: The formula used for fitting the anharmonic phonon decay model has been included in the revised Supplementary Information.

2. Supplementary Fig. 3b has more important information than the current Fig. 1a, such as the loss of surface Cs. The authors may consider revising Fig. 1 to add that information.

Reply: Thank you for your suggestion. We agree that Supplementary Fig. 3b would convey more information on the loss of surface Cs. However, due to the slender crystal structure of the 8L sample, we attempted several arrangements and the plot is not satisfactory. Therefore, we keep Fig. 1a but add a sentence in the caption of Fig. 1: “The schematic structure of an eight-layer thin flake without surface Cs can be found in Supplementary Fig. 3b”. We hope the reviewer will be satisfied with this revision.

3. I cannot see the Raman data from Ref. 57. There is also more data with different thicknesses in Ref. 58's supplement.

Reply: Thank you for your careful reading. The references 57 and 58 in the previous edition were incorrectly cited and have been corrected to references 50 and 51 in the revised version. We have updated the reference.

4. In Fig. 3b, what are those modes near 150 cm^{-1} ?

Reply: Thank you for the comment. We have also noticed the additional phonon modes near 150 cm^{-1} , which appears as a broad hump in the raw data spectra of thin flakes. This phenomenon has also been reported in the supplement of Ref. 50, and should be intrinsic for CsV_3Sb_5 approaching the 2D limit. So far, the origin is unclear. It may be caused by some unknown surface modes and requires further investigations.

5. In Fig. 4, dR/dT in 2-4L v.s. 5L may help to reveal the possible absence of CDW clearly.

Figure R7. dR/dT plot of the 2-5L samples.

Reply: Thank you for your valuable suggestion. As shown in Fig. R7, no anomaly can be observed in the dR/dT plot of the 2 - 4L sample, which is in sharp contrast to the 5L sample. We have included this figure in the Supplementary Information to demonstrate the absence of CDW in the thinner samples.

Reviewer #2:

In this work, the authors report discovery of an unusual enhancement of charge density wave phase in this CsV_3Sb_5 . They observed that the transition temperature to the CDW phase has a minimum for 27 layers and whereas the superconducting transition temperature has a maximum at that thickness. They theorize that this is due to a crossover from electron-phonon coupling (EPC) to dominantly electronic interactions for the CDW phase as the thickness is reduced, but no theoretical model is offered to support this interpretation. This manuscript contains some very interesting data, although the analysis is somewhat incomplete.

Reply: We thank Reviewer #2 for carefully reviewing our paper and providing constructive comments for improving our paper. Our Raman scattering measurements suggest a weakened EPC with decreasing flake thickness. Further combined with the observation of correlated insulating behavior in the atomic-thin flakes and enhanced CDW transition temperature, we propose that the non-monotonic behavior of the CDW transition temperature with the decreasing flake thickness could be attributed to a crossover from EPC to electronic interactions. Both EPC and electronic interactions are believed to be indispensable for the exotic phenomena in these Kagome metals. Our explanation is based on our experimental observation but difficult to be quantitatively described by

a concrete theoretical model due to the complicated landscape of electronic structures in these Kagome metals. Below is our response to the reviewer's technical comments.

1. In page 7, the authors describe the temperature dependence of the field-dependent ρ_{xy} curves, but I cannot find such data in the manuscript of the SI. Please include the actual experimental data.

Reply: We thank the reviewer for this comment. We have included the raw data of ρ_{xy} in the revised manuscript as Supplementary Fig. 6.

Figure R8. Temperature dependence of Hall resistivity. a-c Temperature-dependent Hall resistivity of the 155L, 15L and 7L samples, respectively. The skew resulting from the anomalous Hall effect is located at low fields, while the normal Hall resistivity is linear at high magnetic fields in all samples. The normal Hall coefficients change sign as temperature increase to around 40 K. Meanwhile, the AHE remains to exist up to higher temperatures.

2. In page 7, the authors claim “These observations suggest that the intrinsic property of the CDW order, i.e., time-reversal symmetry breaking, always occurs in thin flakes and survives even approaching the 2D limit.” How is it different from the bulk case, and why?

Reply: We thank the Reviewer #2 for the comment. Actually, the observed AHE effect inside the CDW phase resembles the bulk case and the AHE effect is one of the supporting evidences for the time-reversal symmetry (TRS) breaking of the CDW order. This suggests that the nature of the CDW order in the flakes and bulk is very similar. This claim is highly nontrivial especially considering that the bulk CDW order is 3D with a $2 \times 2 \times 2$ or $2 \times 2 \times 4$ real-space reconstruction [Phys. Rev. X **11**, 031026 (2021), Phys. Rev. X **11**, 041030 (2021)], to which the TRS breaking may be related. The decreasing thickness can have a dramatic effect on the CDW order and the TRS-breaking nature could change. However, our observation suggests that the TRS breaking in the CDW order persists approaching the 2D limit even though the CDW real-space pattern may change.

3. Although the Raman measurements are the most important part of the work, no Raman spectrum is included. The Raman spectra for different thicknesses and their temperature dependence should be presented explicitly. Without them, one cannot judge the validity of the interpretation. The Raman intensity maps of Fig. 3a and b are not enough.

Reply: Thank you for your comment. The raw Raman spectra (Figs. R9, R10, R12, R13) are now included in the Supplementary Information. The 45-layer sample (Fig. R9b) displays a sharp hardening of the A_{1g} mode, while the width of the A_{1g} mode in the 15-layer sample evolves smoothly. These features are quantified and summarized in Fig. 3d, g, and can also be seen in Fig. 3a, b.

Figure R9. Temperature dependent Raman spectrum of 45L sample. **a** The Raman spectrum for 45L sample at temperature varied from 170 K to 4 K. Only two main lattice peaks can be observed. **b** Detail of evolution of the A_{1g} mode. Below 85 K, the A_{1g} mode hardens sharply.

Figure R10. Temperature dependent Raman spectrum of 15L sample. **a** The Raman spectrum

for 45L sample at temperature varied from 170 K to 4 K. Only two main lattice peaks can be detected.

b Detail of evolution of the A_{1g} mode. The width of the A_{1g} mode evolves smoothly.

4. I wonder why only 2 peaks are observed in the Raman spectra. Because the number of atoms per unit cell is very large, I would expect many phonon modes. The authors should include group theoretical analysis of phonon modes and show how many phonon modes are Raman active.

Reply: We thank the reviewer for the comment. From the group-theoretical considerations, phonon modes of the high-temperature phase at the Γ -point can be expressed as $\Gamma_{\text{total}} = A_{1g} + 4A_{2u} + B_{1g} + B_{1u} + 2B_{2u} + 2E_{2u} + E_{2g} + 5E_{1u} + E_{1g}$, among which the Raman-active modes are $\Gamma_{\text{Raman}} = A_{1g} + E_{2g} + E_{1g}$. In the back-scattering geometry used in our experiment, the A_{1g} and E_{2g} can be detected in our *ab*-plane measurement, while the E_{1g} phonon can only be detected from the *ac* plane.

In the CDW phase, the structure is extending to a $2 \times 2 \times 1$ supercell, including two types of structures: Star of David (SoD) and inverse Star of David (iSoD). The space group of these two superlattices is $P6/mmm$, same as the high-temperature phase. Phonon modes at the Γ -point can be expressed as $\Gamma_{\text{total}} = 5A_{1g} + A_{1u} + 3A_{2g} + 9A_{2u} + 4B_{1g} + 5B_{1u} + 2B_{2g} + 7B_{2u} + 8E_{2u} + 8E_{2g} + 14E_{1u} + 6E_{1g}$. Raman-active modes are $\Gamma_{\text{Raman}} = 5A_{1g} + 8E_{2g} + 6E_{1g}$. Clearly, compared to the high-temperature phase, new CDW-induced folding modes will emerge. However, the CDW-folding modes are usually difficult to detect as discussed in the following.

Our temperature-dependent Raman scattering measurements involve both high-temperature and low-temperature phases. At high temperature, only two Raman modes (A_{1g} , E_{2g}) can be observed for both bulk and flake samples, consistent with the group analysis mentioned above and the previous study [Ref. 50]. After the CDW phase transition, seven new folding/amplitude modes ($4A_{1g}$, $3E_{2g}$) emerge and all of them can be observed in the bulk [Refs. 50, 51]. For the thin flakes, the CDW-induced folding modes are hard to be detected because of the low Raman intensity [Ref. 50]. For example, the intensity of the main peak in the 58L sample is only 1/11 of that in the bulk, drowning its weak amplitude/folding modes in the background noise. Therefore, the evolution of the A_{1g} mode, serves as a suitable detector to monitor the evolution of CDW in thin flakes. Based on the evolution of their amplitudes and frequencies, we infer the change of EPC in these Kagome flakes. We have incorporated these analyses in the revised Supplementary Information.

5. The authors claim that the dramatic increase of the CDW transition temperature in thin samples should have a non-phonon origin. Then what is the origin? Can one offer some reasonable explanation?

Reply: We thank the reviewer for the comment. Currently, the origin of the CDW order in the Kagome metals is controversial and there are two scenarios: electron-electron correlation (Peierls transition) and electron-phonon coupling (EPC). In the bulk system, there are accumulating evidences of EPC. However, our Raman scattering measurements suggest a weakened EPC with decreasing thickness, which cannot account for an enhanced CDW transition temperature (below 25 L) in experiments. Due to generally enhanced correlation effect approaching 2D and intrinsic

van Hove singularities near the Fermi level in these Kagome metals, we propose that the electron-electron interaction can further promote the CDW and enhance its transition temperature in the atomic-thin flakes. This is supported by recent papers [Nat. Commun. 13, 6348 (2022), arXiv:2210.16890] reporting the $2\times 2\times 2$ to $2\times 2\times 1$ transition by applying pressure or reducing the sample thickness.

6. The authors used 2 mW of the laser power focused to a micron-sized spot. This corresponds to a very high intensity, which may well damage the sample. Is there any data that shows that the laser did not cause damage to the specimen?

Reply: The 2 mW is the initial laser power. Our samples are mounted inside a closed temperature-controlling stage (4 - 300 K). When the laser passes through the optical window, the intensity is significantly reduced to ~ 0.4 mW. Figure R11 shows the optical images before and after the experiment and no damage can be seen on the sample surface. Additionally, the Raman spectrum obtained by long-time collection is consistent with the literature reports, which further indicates that the current parameters will not cause damage to the sample. To avoid confusion, we have deleted the value of initial power in the revised manuscript.

Figure R11. Optical images before and after the experiment. The laser power is 2 mW, and the exposure time is 1800 s.

7. In page 7, the authors refer to Fig. 3d, which should be Fig. 2d.

Reply: Thank you for pointing this out. We have corrected it in the revised manuscript.

Reviewer #3:

Report on the manuscript entitled “Anomalous enhancement of charge density wave in kagome superconductor CsV_3Sb_5 approaching the 2D limit”.

The manuscript reports the study of the Cs-based Kagome superconductors family as a function of number of layers down to 2 layers. The authors employed transport and Raman measurements, as well as characterizations such as TEM and X-ray diffraction. This work represents the first comprehensive study on this family in few layers limit and on clean samples. An interesting phase diagram is revealed, with competing CDW and superconductivity but non monotonic behaviour as a function of number of layers. The CDW tends to disappear smoothly down to 20-25L then is strongly reinforced below this value. Below 4L, the system becomes insulator and of course is no

more superconductor. The authors claimed that this phase diagram can be accounted by a change of dominant mechanism in the CDW drivers, from EPC to electronically driven.

This study is new and well accomplished, it thus deserves publication as soon as possible. Nevertheless I have some remarks.

Reply: We thank Reviewer #3 for his/her careful reviewing of our paper and we appreciate his/her positive and constructive comments for improving our paper. Below is our response to his/her comments and concerns.

First, the Raman results are limited to the two high energy phonons that are already present in the normal phase. The previous Raman studies on bulk system have shown clear new peaks and amplitudons in the CDW. The authors must look for these features, and comment on them. If their apparatus allows it, they can also look for low energy excitations like quasi-elastic line and mode below 50 cm^{-1} . If they have no access to this, many modes are also expected above 70 cm^{-1} . I then strongly encourage to show these results or to perform more Raman experiments down to 2L on the present samples. I also encourage to show the raw spectra in the Supplementary in addition to the color plots.

Figure R12. CDW-induced modes in bulk CsV_3Sb_5 . **a** XX configuration measurement in bulk CsV_3Sb_5 . Dashed lines show the main lattice modes. The dotted lines show CDW-induced modes. **b** Lorentz fitting of the 100 K Raman spectrum in **a**. The black, green, blue, and red lines are raw data, baseline, fitted peaks, and fitted line, respectively.

Figure R13. Raman spectra for bulk and thin flake (58L) at 4 K. The laser power and collection time are set to be the same.

Reply: We thank the reviewer for the informative comments and nice suggestion. We have carried out measurements of the Raman spectrum down to 50 cm^{-1} . Figure R12 shows the raw spectrum of bulk CsV_3Sb_5 with an exposure time of 1800 s. At 100 K, only two peaks, A_{1g} and E_{2g} , are detected. At 4 K, seven new peaks emerge in the Raman spectrum, agreeing well with the previous publication [Ref. 50]. Under the XX configuration used in our experiments, A_{1g} and E_{2g} modes are collected simultaneously. We then use an exposure time of 600 s to measure the bulk sample. As shown in Fig. R13, the intensity of CDW folding modes of the bulk sample (red) decreases to about 1/3 of that in Fig. R12. The general feature of the CDW folding modes can still be seen clearly except that two peaks in 197 cm^{-1} and 205 cm^{-1} merge to a hump. To trace the CDW-induced folding modes in thin flakes, we further measured a relatively thick flake (58L) under the same laser power and exposure time (600 s) for comparison. However, no trace of the amplitude/folding peak can be detected even in the 58L flake (blue curve in Fig. R13), which is in agreement with a recent publication [Ref. 50] that the folding modes cannot be distinguished below 68L. Therefore, in our manuscript, the evolution of the A_{1g} mode, serves as a suitable detector to monitor the evolution of CDW in thin flakes. We have included the raw Raman spectra in the revised Supplementary Information.

Secondly, I would be interested to see the transport data without shift. Indeed, the comparison of the Temperature tendency, power-law behavior ... as a function of number of layers might help to conclude on CDW gap or other dimension-limited behavior. More information can be extracted from the present transport studies.

Figure R14. Resistance of CsV_3Sb_5 thin flakes. **a** Raw data of resistance without normalization or shifting. The thicknesses of samples are corresponding to the line with the same color in the main text Fig. 2a. **b-i** Power-law fitting of low-temperature resistance.

Reply: We thank Reviewer #3 for the nice suggestion. Figure R14 shows the raw transport data without any normalization or shifting. Power-law fitting of the low-temperature normal state resistance gives almost comparable parameter α across the entire range of sample thicknesses. Interestingly, the linear fitting of the high-temperature resistance before entering the CDW order shows a sharp drop of the slope crossing the critical thickness of 25L, as shown in Fig. R15, indicating its close relationship with the anomalous enhanced T_{CDW} . We have included this information in the revised manuscript.

Figure R15. Linear fitting of thickness-dependent resistance above T_{CDW} . **a** Normalized resistance of samples in normal metal phase. **b** Slope extracted from linear fitting of data in **a**. The dashed line indicates the critical thickness of 25L.

Here are additional comments:

P7: the link between TRSB and CDW (through AHE) should be discussed more and clarified.

Reply: We thank the reviewer for the comment. Generally, the AHE occurs in the system with broken TRS. When TRS is broken, it opens a gap at the Dirac point near the M point in CsV_3Sb_5 , resulting in a large Berry phase and giving rise to the intrinsic AHE. In the bulk, the AHE inside the CDW phase is believed to be one of the evidences supporting the TRS breaking of the CDW order, which is further confirmed by the recent μ SR [Ref. 40] and MOKE [Ref. 44] measurements. In the thin flakes, we also observed AHE inside the CDW order, suggesting that the CDW order in both bulk and thin flakes breaks TRS. We have modified the corresponding sentences in the main text to make it clearer.

P9: the authors said: “The gradual decline of the EPC with decreasing thickness can account for the suppression of CDW order at above 25L”. Please clarify what you mean (“at above”).

Reply: We thank the reviewer for this comment. We mean the flakes with a thickness larger than 25 layers. We modify this to “... account for the suppression of CDW order above 25L”

P9: the authors said: “the absence of CDW kink and superconductivity may be ascribed to the loss of surface Cs by exfoliation.” What do you mean? Do you think this is an effect of change in doping level? Do you know the doping expected from the Cs layer?

Reply: Actually, we mean the charge doping effect from the loss of surface Cs atoms. In the atomic-thin flakes, this type of charge doping can be much more significant than the thick flakes, which can dramatically change the electronic properties. We modify this to “... may be ascribed to the charge doping from the loss of surface Cs by exfoliation.” We estimate the doping content by measuring the thickness-dependent Hall effect, as shown in Fig. R2. The extracted hole-carrier density at 4 K shows a sharp increase when approaching the 2D limit. We have incorporated this information into the revised manuscript.

P9: The author fitted the resistivity at low temperature. The conclusion for the presence of a coulomb gap may be softened as the fit concerns only 1K range. The fitting alone doesn't prove the existence of Coulomb gap (a gap yes but the mechanism is not proven).

Reply: We agree with Reviewer #3. Apart from the Coulomb repulsion, other possibilities such as crystalline imperfections approaching the atomic limit, buckling due to 2D fluctuations, and percolation through grain boundaries can also produce an Anderson-like variable-range-hopping behavior. We have discussed these possibilities in the manuscript.

Could the authors clarify the difference between figure S3c and d?

Reply: Thank you for your careful reading. Figure S3c displays the experimental high-angle annular dark-field scanning transmission electron microscopy (HAADF-STEM) image of an 8L sample. The simulated atomic resolution image, based on the crystal structure model shown in Fig. S3b, is presented in Fig. S3d and was generated using the xHREMTM software. We have revised the caption to make it clearer.

The authors mentioned that Fig S6 presents some dependence in the main but there is no temperature dependence in Fig. S6.

Reply: We thank the reviewer for this comment. The temperature-dependent ρ_{xy} has now been included in the Supplementary Information as Supplementary Fig. 6.

To conclude, this work is very interesting, new and will be of interest for a large community in condensed matter. At present, the analysis and maybe Raman measurements were not used as much as the authors could have done. More information can be extracted from this study. I then strongly encourage to improve this before publication in Nature Communication.

Reply: We thank the reviewer for the positive comments and potential recommendation.

Revision list:

1. In page 7, we added three sentences “Generally, the AHE...”
2. In page 12, we included a paragraph “The power-law...”
3. We added two data points into Fig. 3i.
4. We included the temperature dependence of Hall resistivity into Supplementary as Supplementary Fig. 6.
5. We included carrier density and mobility at 4 K as Supplementary Fig. 7.
6. We included temperature dependent Raman spectrum of 45L sample as Supplementary Fig. 9.
7. We included temperature dependent Raman spectrum of 15L sample as Supplementary Fig. 10.
8. We included CDW-induced modes in bulk CsV₃Sb₅ as Supplementary Fig. 11.
9. We included CDW-induced modes in CsV₃Sb₅ thin flakes as Supplementary Fig. 12.
10. We included group analysis as Supplementary Note 7.
11. We included anharmonic phonon decay model as Supplementary Note 8.
12. We included temperature-dependent Raman response of 28L and 20L samples measured in the XX configuration as Supplementary Fig. 13.
13. We included the sensitivity of A_{1g} mode as Supplementary Note 9.
14. We included dR/dT plot of the 2-5L samples as Supplementary Fig. 14.
15. We investigated the influence of evaporated-Al₂O₃ layer as Supplementary Fig. 16.
16. We included resistance of CsV₃Sb₅ thin flakes as Supplementary Fig. 17.
17. We included linear fitting of resistance above T_{CDW} as Supplementary Fig. 18.
18. We included superconducting transition width as Supplementary Fig. 19.
19. We included analysis of transport data as Supplementary Note 11.

REVIEWER COMMENTS

Reviewer #1 (Remarks to the Author):

This is my second read of this manuscript by Boqin Song, et al. In the last round of review, the main technical concern is the interpretation and quality of Raman data. The author addressed it very well with additional measurements and analyses.

The authors very well incorporated all suggestions and improved the manuscript substantially. The manuscript is now ready for publication in Nature Communications.

Reviewer #2 (Remarks to the Author):

The authors have revised the manuscript and SI to address the comments by the reviewers. Although not all my questions are answered, I think the authors' explanations are acceptable. Given that this is a relatively new material system, it is reasonable that not all details are explained. I recommend that this manuscript be accepted without further review.

Reviewer #3 (Remarks to the Author):

Report n°2 on the manuscript entitled "Anomalous enhancement of charge density wave in kagome superconductor CsV3Sb5 approaching the 2D limit"

The authors replied to most of my comments and questions. However there is still an important question that must be addressed before publication. It concerns the Raman data.

Indeed, regarding the Raman results, what is the difference between data acquisition process between R12 and R13. Is it only acquisition time? If so, there is no reason to reduce the acquisition time as the authors lower the signal to noise ratio.

Generally we need a clear comparison of the Raman signal between different samples with different thickness. I suggest to scale the data of figure R13 on the intensity of the mode at 120 cm⁻¹ data (after removing the dark signal). Then we can compare the noise in thin sample with the CDW folded mode in the bulk. Presently Fig R13 presents data on the 58L that are difficult to judge.

Generally I suggest to present data as cnts/s.mW. And if there is a strong loss of signal on thin samples, to scale them all on the highest intense plain phonon mode.

Most importantly, it is really worth trying to measure the folded modes, as these are major signature of CDW and of electron-phonon coupling. To access to them, and since 600s acquisition is not so long, the authors can look more carefully for signature of CDW folded modes with much longer acquisition time (or many acquisitions and average) on thin samples.

The authors mentioned an impossibility to measure folded modes below 68L, with ref [50].

Ref 50 is this paper: Liu, G. et al. Observation of anomalous amplitude modes in the kagome metalCsV3Sb5. Nat. Commun. 13, 3461 (2022).

There is no mention of 68L sample in this paper. Even in the supplementary part, they measured a 68nm sample and others below but they observed a loss of the main phonons (not folded ones) when reducing thickness. So there is no general conclusion on such impossibility to measure folded modes.

Response to Reviewers

We thank Reviewers #1 and #2 for recommending the publication of our work without further review. We appreciate Reviewer #3 for the valuable suggestion to perform a detailed analysis and further experiments on the Raman spectra. To address Reviewer #3's concerns and comments, we have tried our best to perform additional Raman measurements and have successfully observed the CDW-induced folding modes in the thin flakes. We greatly appreciate all Reviewers for helping us to improve our work.

Reviewer #1 (Remarks to the Author):

This is my second read of this manuscript by Boqin Song, et al. In the last round of review, the main technical concern is the interpretation and quality of Raman data. The author addressed it very well with additional measurements and analyses.

The authors very well incorporated all suggestions and improved the manuscript substantially. The manuscript is now ready for publication in Nature Communications.

Reply: We thank Reviewer #1 for recommending the publication of our work.

Reviewer #2 (Remarks to the Author):

The authors have revised the manuscript and SI to address the comments by the reviewers. Although not all my questions are answered, I think the authors' explanations are acceptable. Given that this is a relatively new material system, it is reasonable that not all details are explained. I recommend that this manuscript be accepted without further review.

Reply: We thank Reviewer #2 for recommending the publication of our work.

Reviewer #3 (Remarks to the Author):

Report n°2 on the manuscript entitled “Anomalous enhancement of charge density wave in kagome superconductor CsV₃Sb₅ approaching the 2D limit”. The authors replied to most of my comments and questions. However, there is still an important question that must be addressed before publication. It concerns the Raman data.

Indeed, regarding the Raman results, what is the difference between data acquisition process between R12 and R13. Is it only acquisition time? If so, there is no reason to reduce the acquisition time as the authors lower the signal to noise ratio.

Reply: We appreciate Reviewer #3 for the valuable comment. The difference between R12 and R13 is only the acquisition time. We agree with your critique and adopt a much longer acquisition time in the following additional experiments. Previous Figure R13 with a shorter acquisition time has been removed from the SI.

Generally we need a clear comparison of the Raman signal between different samples with different

thickness. I suggest to scale the data of figure R13 on the intensity of the mode at 120 cm⁻¹ data (after removing the dark signal). Then we can compare the noise in thin sample with the CDW folded mode in the bulk. Presently Fig R13 presents data on the 58L that are difficult to judge. Generally I suggest to present data as cnts/s.mW. And if there is a strong loss of signal on thin samples, to scale them all on the highest intense plain phonon mode.

Most importantly, it is really worth trying to measure the folded modes, as these are major signature of CDW and of electron-phonon coupling. To access to them, and since 600s acquisition is not so long, the authors can look more carefully for signature of CDW folded modes with much longer acquisition time (or many acquisitions and average) on thin samples.

The authors mentioned an impossibility to measure folded modes below 68L, with ref [50]. Ref 50 is this paper: Liu, G. et al. Observation of anomalous amplitude modes in the kagome metal CsV₃Sb₅. Nat. Commun. 13, 3461 (2022). There is no mention of 68L sample in this paper. Even in the supplementary part, they measured a 68nm sample and others below but they observed a loss of the main phonons (not folded ones) when reducing thickness. So there is no general conclusion on such impossibility to measure folded modes.

Reply: We thank Reviewer #3 for the valuable comments and suggestions. To visualize the CDW-induced folding modes, we extend the acquisition time for each round of measurement to 1 hour (3600 s) in the additional Raman measurements. Furthermore, we notice that the intensity of these folding modes decreases rapidly in thin flakes. To address this, we follow Reviewer #3's advice and repeat the measurement multiple times, taking the average to enhance the signal-to-noise ratio. We then scale the spectra to the highest phonon mode (E_{2g}) at 117 cm⁻¹, and all data are presented in cnts·mw⁻¹·s⁻¹ units.

The Raman spectra for different sample thickness (695L, 96L, 48L, 37L, 22L, 16L) are shown in Figure R1. After scaling, the CDW-induced folding modes are clearly distinguishable in thin flakes with thickness down to 37L, which is consistent with the observed modes in the bulk. We have made several attempts to measure the folding modes on thinner flakes (22L, 16L) by prolonging the acquisition time and conducting multiple repetitions. But regrettably, we are unable to detect it. We suspect that this could be related to the critical thickness of 25L, although the underlying mechanism remains unclear.

Therefore, our previous claim that detecting folding modes below 68L is impossible is thus incorrect. We have removed the relevant content from the manuscript and include Fig. R1 in the SI. We thank Reviewer #3 again for these excellent suggestions, and hope this revision resolves the concerns.

Figure R1. Raman spectra of 695L, 96L, 48L, 37L, 22L and 16L flakes collected at 4 K. All the spectra have been scaled using the intensity of the E_{2g} phonon mode at 117 cm^{-1} and vertically shifted for clarity.

Revision list:

1. In page 8, we modified a sentence to “While the folding modes...”
2. We removed the Supplementary Fig. 12 in the previous version and included above Fig. R1 as the new Supplementary Fig. 12 in the revised SI.

REVIEWERS' COMMENTS

Reviewer #3 (Remarks to the Author):

Report 3 on the manuscript entitled "Anomalous enhancement of charge density wave in kagome superconductor CsV₃Sb₅ approaching the 2D limit"

The authors improved the Raman measurements and were able to observe the folded phonon modes of the CDW down to 37K. Of course that would be interesting to follow their temperature/number of layers dependences but regarding the novelty of the topic, I am happy to accept the manuscript for publication in Nature Com. in the present status.